# The Role of *Energy Homeostasis-Associated* Gene Expression and Serum Adropin Levels in Patients with Familial Mediterranean Fever

**DOI:** 10.3390/ijms26052371

**Published:** 2025-03-06

**Authors:** Durkadin Demir Eksi, Gulay Gulbol Duran, Muhammet Murat Celik, Yunus Emre Eksi, Ramazan Gunesacar

**Affiliations:** 1Department of Medical Biology, Faculty of Medicine, Alanya Alaaddin Keykubat University, Antalya 07425, Türkiye; durkadin.eksi@alanya.edu.tr (D.D.E.); yunus.eksi@alanya.edu.tr (Y.E.E.); 2Department of Medical Biology, Tayfur Ata Sökmen Faculty of Medicine, Hatay Mustafa Kemal University, Hatay 31001, Türkiye; gulayduran@gmail.com; 3Department of Internal Medicine, Tayfur Ata Sökmen Faculty of Medicine, Hatay Mustafa Kemal University, Hatay 31001, Türkiye; muratcelikdr@yahoo.com; 4Department of Internal Medicine, Faculty of Medicine, Siirt University, Siirt 56100, Türkiye

**Keywords:** familial mediterranean fever (FMF), adropin, ENHO

## Abstract

Familial Mediterranean Fever (FMF) is a genetic autoinflammatory disease primarily affecting populations in the Mediterranean region. The pathogenesis of FMF and the roles of various molecules remain unclear. Adropin, a protein encoded by the Energy Homeostasis-Associated Gene (*ENHO*), is involved in energy metabolism and inflammation. This study aimed to explore the relationship between *ENHO* expression, Adropin levels, and FMF, examining their correlations with disease characteristics. This study included 30 patients clinically diagnosed with FMF and 35 healthy controls. The *ENHO* expression in peripheral blood mononuclear cells was assessed using a qRT-PCR, and the serum Adropin levels were measured via ELISA. The *ENHO* expression was significantly elevated in the FMF patients compared to the controls (*p* = 0.0007), while no significant differences were observed in the serum Adropin levels between the groups (*p* = 0.81). A correlation analysis revealed a negative association between the *ENHO* expression and age (r = −0.47, *p* = 0.009), whereas the serum Adropin levels were positively correlated with age, disease onset, and diagnostic delay (*p* < 0.05). No significant associations were found between the *ENHO* expression and Adropin levels or FMF clinical features. These findings suggest that increased *ENHO* expression may play a role in FMF pathophysiology, potentially as a compensatory mechanism. The correlation between Adropin levels and disease onset indicates a potential protective role. Further studies are needed to confirm these findings.

## 1. Introduction

Familial Mediterranean Fever (FMF) (OMIM#2149100) is an autoinflammatory disease characterized by recurrent short episodes of peritonitis, pleuritis, arthritis, and rash, and is often accompanied by a fever. FMF attacks usually begin in early childhood, and 80–90% of patients become symptomatic before the age of 20 [1]. FMF is the most common autoinflammatory disease and is also the most prevalent of the monogenic periodic fever syndromes. Although it primarily affects people living in the Eastern Mediterranean region, it is now seen worldwide due to travel and migration since the twentieth century [2]. It is most commonly seen among Turks, Armenians, Jews, and Arabs. The prevalence of FMF in endemic countries ranges between 1 in 500 and 1 in 1000, with the highest reported prevalence of 1 in 395 in the Central Anatolia region of Türkiye [1]. The two most significant milestones in the history of FMF were the initiation of colchicine as a treatment in 1972 and the identification of mutations in the Pyrin (Innate Immunity Regulator, *MEFV*) gene as the cause of FMF in 1997 [2]. FMF is generally inherited in an autosomal recessive manner, though some heterozygotes have been found to exhibit a clinical spectrum ranging from mild to classic FMF symptoms [3]. Recent studies have indicated that the inheritance pattern of FMF deviates from the conventional autosomal recessive model typical of monogenic Mendelian disorders, with evidence suggesting a more intricate mechanism of penetrance and pathogenesis [2].

Adropin is a secretory molecule encoded by the Energy Homeostasis-Associated Gene (*ENHO*) that plays an active role in energy homeostasis. The *ENHO* is expressed in various organs and tissues, including endothelial cells, as well as the liver, brain, pancreas, heart, muscles, umbilical vein, and kidneys. Adropin is involved in the regulation of energy balance, insulin sensitivity, and the metabolism of glucose and lipids. In addition to its metabolic functions, Adropin also mediates non-metabolic processes, including angiogenesis, apoptosis, and inflammation [4]. The association of Adropin with inflammation and vasculopathy suggests its potential involvement in inflammatory diseases. Previous studies have investigated serum Adropin levels and *ENHO* expression in patients with systemic sclerosis, Behçet’s disease, rheumatoid arthritis (RA), and systemic lupus erythematosus [4,5]. FMF is a condition with an incompletely understood pathogenesis, which poses challenges for both its accurate diagnosis and effective treatment [6]. Although colchicine is the standard therapy for FMF, around one-third of patients achieve only partial remission, while approximately 5–10% do not respond to treatment [7]. Aside from an *MEFV* analysis, there are no specific laboratory tests available to substantiate the diagnosis of FMF. No genetic mutations can be detected in approximately 30–40% of patients with FMF [6].

Various studies have demonstrated that endothelial dysfunction is one of the complications associated with FMF [8,9]. On the other hand, the interplay between the immune system and metabolism plays a significant role in the development of autoimmune diseases. It has been observed that individuals with autoinflammatory diseases have a higher prevalence of metabolic syndrome [10]. Considering these factors, we aimed to investigate serum Adropin levels and *ENHO* expression in clinically diagnosed FMF patients, based on the hypothesis that Adropin may influence the clinical course of FMF.

## 2. Results

### 2.1. Clinical Findings

This study included 30 patients clinically diagnosed with FMF (16 male, 14 female) with a mean age of 34.33 (13.79) years. A control group comprising 35 age- and sex-matched healthy individuals was also enrolled. The patients were classified as ’active’ during physical examinations and blood sampling if they were experiencing active episodes, while those in attack-free periods were classified as ‘inactive’. Six out of thirty patients (20%) were active, and the rest were inactive. The demographic data, comprehensive clinical findings, biochemical parameters, and colchicine dosages of the FMF patients are summarized in Table 1.

### 2.2. ENHO Expression Profiles

The ∆Ct values for the *ENHO* expression in both the control and patient groups were calculated by normalizing them to the *GAPDH* expression levels. A comparison of the relative *ENHO* expression between the groups using the 2^−ΔCt^ method revealed that the patient group had a significantly higher expression level compared to the control group (*p* = 0.0007) (Figure 1). The calculation of the 2^−∆∆Ct^ value indicated that *ENHO* expression in the patient group was elevated by 3.72-fold relative to the control group.

### 2.3. Serum Adropin Levels

No statistically significant difference was observed in the serum Adropin levels between the patient and control groups (58.22 (41.51) pg/mL and 56.83 (43.42) pg/mL, respectively; *p* = 0.81; the data are presented as the mean (SD)) (Figure 2).

### 2.4. Evaluation of Clinical, Biochemical, and Gene Expression Findings of Patients with FMF

The clinical, biochemical, and *ENHO* expression data were analyzed and compared in the FMF patients based on the presence of specific disease symptoms. The number of attacks per month in patients with abdominal pain has been found to be significantly higher compared to those without (*p* = 0.04). The diastolic blood pressure (DBP) values for the patients experiencing abdominal pain and chest pain were significantly elevated compared to those without these symptoms (*p* = 0.01 and *p* = 0.04, respectively). Elevated microproteinuria levels were also found in the patients with chest pain compared to those without (*p* = 0.03). Additionally, the FMF patients without scrotal pain had statistically higher ESR levels compared to those with scrotal pain (*p* = 0.04). No significant differences were observed for the other parameters between the groups (Table 2).

The laboratory parameters, *ENHO* expression, and Adropin levels of the active and inactive patients were compared. Accordingly, the C-reactive protein (CRP) and fibrinogen levels were found to be significantly higher in the active patients compared to the inactive patients (*p* < 0.001 and *p* = 0.036, respectively). On the other hand, the DBP was significantly higher in the inactive patients compared to the active patients (*p* = 0.004). No statistical differences were observed between the active and inactive patients in terms of the *ENHO* expression, Adropin levels, or other biochemical parameters (Table 3).

### 2.5. Correlations Between FMF Characteristics and Laboratory Findings of Patients

A Spearman correlation analysis revealed a negative correlation between the *ENHO* expression level and the age of the patients (r = −0.47; *p* = 0.009; 95% CI: −0.7149–0.1199). No significant correlation was found between the *ENHO* expression and any other clinical parameters (*p* > 0.05) (Figure 3).

A positive correlation was also observed between the serum Adropin levels and the patients’ age (r = 0.39; *p* = 0.033; 95% CI = 0.0229–0.6638). Furthermore, the serum Adropin levels exhibited a positive correlation with disease onset (r = 0.45; *p* = 0.014; 95% CI = 0.0904 − 0.7075) and the age at FMF diagnosis (r = 0.52; *p* = 0.004; 95% CI = 0.1767 − 0.7488). No significant correlation was found between the serum Adropin levels and any other clinical parameters (*p* > 0.05) (Figure 3).

## 3. Discussion

FMF is the most prevalent monogenic autoinflammatory disorder, with a particularly high incidence among Jewish, Armenian, Turkish, and Arab populations. However, due to global migration, FMF has become a condition that can now be encountered in many regions worldwide (25). The routine diagnostic tests for FMF typically include inflammatory biomarkers, such as neutrophilic leukocytosis (≥20,000/mm³), the erythrocyte sedimentation rate (ESR), CRP, serum amyloid-A, and elevated immunoglobulin levels during acute attacks. In addition, an *MEFV* gene mutation screening is commonly performed, although 30–40% of patients do not present with detectable mutations. In these cases, when the clinical symptoms strongly suggest FMF despite negative genetic testing, a trial of colchicine therapy is often prescribed for 3–6 months. A positive response to colchicine followed by symptom recurrence upon discontinuation can solidify a diagnosis of FMF. This diagnostic strategy involves an open-label approach [6]. Interestingly, even among siblings with identical genetic mutations, the clinical presentation of FMF can vary significantly [11], suggesting the presence of disease-modifying factors, the nature of which remains poorly understood. These observations point to the potential involvement of other genes or signaling pathways, particularly those associated with inflammation, in shaping the FMF phenotype.

Adropin, a secretory protein encoded by the *ENHO*, plays a critical role in regulating energy metabolism and insulin sensitivity [12]. It has been implicated in various conditions, such as multiple sclerosis, COVID-19, gestational diabetes, obstructive sleep apnea, RA, coronary artery ectasia, acute mesenteric ischemia, and diabetic nephropathy [13]. Clinical studies have revealed reduced Adropin expression in inflammatory diseases, alongside a negative correlation with inflammatory cytokine levels, suggesting potential anti-inflammatory effects [12]. The association of the *ENHO* and/or Adropin with other autoimmune diseases has also been investigated. Studies on autoimmune inflammatory diseases have reported both decreases and elevations in Adropin levels, suggesting a complex and context-dependent role of Adropin in these disorders [4,5,14]. In our study, the *ENHO* expression was found to be significantly higher in the clinically diagnosed FMF patients compared to the healthy controls. However, no significant differences were observed in the serum Adropin protein levels between the two groups. A reason for this lack of correlation may have been the inability to evaluate the *ENHO* expression for the entire control group, as some individuals had insufficient RNA quality and quantity. Several factors may underlie the increased *ENHO* expression levels observed in FMF. It is well established that the activation of Liver X Receptor (LXR) signaling reduces *ENHO* expression in the liver [15]. In our study, the upregulation of *ENHO* expression may have been linked to the downregulation of the LXR signaling pathway. LXR activation is known to suppress inflammation and neutrophil migration [16]; therefore, its potential downregulation may contribute to the inflammatory processes in FMF. On the other hand, the increased *ENHO* expression in the FMF patients, which is regulated by LXR and other molecular pathways, may represent a protective adaptive mechanism aimed at suppressing inflammation. This hypothesis is supported by the well-documented anti-inflammatory effects of Adropin [15].

Yolbaş and colleagues found a significant increase in serum Adropin levels in patients with systemic sclerosis and Behçet’s disease compared to healthy controls, although no difference was observed in *ENHO* expression [5]. In a separate study conducted by the same team, *ENHO* expression was significantly elevated in patients with RA, but the serum Adropin levels remained similar between the RA patients and the healthy controls. In the same study, no significant differences were observed in the *ENHO* expression and serum Adropin levels between the patients with systemic lupus erythematosus and the healthy controls [4]. This study shows a similarity to our findings, as the *ENHO* expression was significantly higher in the FMF patients compared to the healthy controls, yet this difference did not correlate with changes in the Adropin levels. This observed discrepancy may be attributed to the post-transcriptional degradation of *ENHO* mRNA in blood tissue or post-translational modifications, leading to the degradation of Adropin. Recent studies, including those by Yolbas et al. [4,5], have demonstrated that *ENHO* mRNA is expressed in peripheral blood mononuclear cells (PBMCs), although at lower levels compared to tissues like the liver and brain. Our study also confirmed measurable *ENHO* expression in the PBMC samples from both the FMF patients and the controls, with significant differences observed between the groups. These findings support the use of PBMCs as a valid target tissue for investigating *ENHO* expression in blood-derived cells.

In our study, the number of attacks (per month) was found to be significantly higher in patients with abdominal pain (*p* = 0.04). It has been reported that 80–90% of patients experiencing attacks also have abdominal pain associated with peritonitis, as observed in our patient group [17]. Therefore, it was expected that the patients with abdominal pain would have a higher number of attacks. In the FMF patients with abdominal and chest pain, the DBP was recorded to be significantly higher compared to those without such symptoms (*p* = 0.01 and *p* = 0.04, respectively). The DBP values were also significantly elevated in the active FMF patients (*p* = 0.004). Although there were statistical differences between these groups in terms of the DBP, the DBP remained within the physiological limits for all the groups. It is known that due to this persistent and recurring inflammatory condition, patients may be at an elevated risk of cardiovascular events [18]. Additionally, the FMF patients with scrotal pain showed significantly lower ESR levels compared to those without scrotal pain. In many cases, inflammation remains active even during periods without attacks. This subclinical inflammation exposes patients to a higher likelihood of developing different complications [19]. Microproteinuria levels were found to be elevated in our patients with localized pain, except in those with scrotal pain. In the patients with chest pain, the microproteinuria levels were significantly higher compared to those without chest pain (*p* = 0.03). The patients in our study responded positively to the colchicine treatment, which may explain the absence of amyloidosis as a clinical manifestation. Given the known association between colchicine responsiveness and a reduced risk of amyloidosis [20], this outcome is consistent with previous findings for FMF patients.

In our FMF patients who had blood samples taken during an attack (active patients), the CRP and fibrinogen levels were found to be significantly higher compared to the inactive patients (*p* = 0.036 and *p* < 0.001, respectively), while the DBP measurements were lower (*p* = 0.004). The acute-phase inflammatory biomarker CRP was found to be elevated in the active patients, reflecting the heightened inflammatory state of this group. Fibrinogen is an important molecule in regulating the inflammatory response. One study reported the highest fibrinogen levels in FMF patients during acute attacks, intermediate levels in those who were attack-free, and the lowest levels in healthy controls [21]. Our finding is consistent with the results of this study. Fibrinogen is known to be upregulated by inflammatory mediators. Elevated fibrinogen levels have been observed during the acute phases of inflammatory/autoimmune diseases, such as inflammatory bowel disease, RA, multiple sclerosis, and vasculitis [22]. It has been reported that during inflammation, vascular permeability increases, and additionally, the rise in fibrinogen levels further enhances vascular permeability. The increase in vascular permeability triggers the extravasation of fibrinogen [23]. No associations were found between the serum Adropin levels, *ENHO* expression, and the clinical findings of the patients. However, these findings need to be confirmed in larger patient cohorts to better understand the underlying mechanisms and their clinical implications.

The pathogenesis of FMF has not yet been fully elucidated. Alternative pathways involving the immune system and microbiota have been proposed in the development of autoinflammatory diseases. Murdaca et al. highlighted the interaction between IL-33 and IL-31 in amplifying inflammatory cascades [24], which may contribute to the systemic inflammation observed in FMF patients, particularly during acute attacks. Moreover, the intricate interplay of Th17 cells and their signature cytokines, such as IL-17, has been implicated in the pathogenesis of numerous chronic inflammatory and autoimmune diseases [25]. FMF, characterized by episodic systemic inflammation, shares overlapping mechanisms with these disorders. Additionally, vitamin D deficiency, intestinal dysbiosis, and their interactions with aging and sex hormones have been implicated as modulators of inflammation and immune responses in autoimmune and inflammatory diseases [26]. Given the shared inflammatory mechanisms, these factors could potentially play a role in FMF pathophysiology as well, warranting further investigation. As Adropin is generally known as an anti-inflammatory protein, future studies should be conducted to explore the potential association between *ENHO* expression and these biological mechanisms.

In the current study, a positive correlation was found between the patients’ age, disease onset, age at diagnosis, and diagnostic delay. This is an expected and explainable result. However, the negative correlation between the patients’ age and *ENHO* expression, along with the positive correlation between age and serum Adropin levels, is intriguing. These findings may suggest a complex interaction between aging, inflammatory processes, and metabolic regulation. Aging could influence immune responses, affecting *ENHO* expression, while changes in metabolic and inflammatory pathways could impact Adropin levels. Further investigation is needed to explore these mechanisms. Additionally, a positive correlation was observed between disease onset, age at diagnosis, diagnostic delay, and serum Adropin levels. The onset of FMF symptoms varies, with 65% of cases presenting within the first decade of life and 90% within the first 20 years [27]. In children, the delayed onset of disease-associated symptoms has been associated with a lower frequency of annual attacks, a milder disease progression, and a reduced colchicine dosage during treatment [28]. We also analyzed the relationship between the patients’ age and their clinical characteristics. But, no significant correlation was found between the patients’ age, disease onset, and clinical findings. However, there was a positive correlation observed between the circulating Adropin levels and disease onset, meaning that higher Adropin levels were associated with a later onset of FMF, suggests that Adropin may delay the onset of FMF symptoms and could potentially play a protective role. On the other hand, previous studies have shown that circulating Adropin levels tend to decrease with advanced age in patients with various phenotypes of the disease [29,30].

Our study is the first to investigate the relationship between *ENHO* expression, circulating Adropin levels, and the clinical characteristics of FMF. The significantly increased expression of the *ENHO* in the clinically diagnosed FMF patients suggests that this gene may play a role in the pathogenesis of the disease. However, the absence of a significant difference in the serum Adropin levels between the patient and control groups may indicate that Adropin exerts its effects primarily at the cellular or local level rather than systemically. Additionally, the positive correlation between disease onset and Adropin levels in the FMF patients suggests a protective role of Adropin in the disease. The limitations of our study include the relatively small sample size and the lack of *MEFV* gene sequencing data for the patients. The distribution of *MEFV* gene variants varies across populations [31,32]. There is evidence suggesting that these variations contribute to the diversity of FMF clinical manifestations. In one study, a genotype–phenotype analysis revealed a more severe disease course in FMF patients with M694V or V726A mutations compared to those with E148Q homozygosity, both of which were among the most frequently identified mutations in FMF patients [33]. A large-scale study conducted in Turkey showed that patients with the M694V/M694V genotype had an earlier age of onset and a higher frequency of arthritis and arthralgia than other groups [34]. Therefore, *MEFV* genotyping could play an important role in improving disease management and guiding genetic counseling [33]. Further research with larger patient cohorts, incorporating both phenotypic and genotypic evaluations, would help elucidate the role of Adropin in the development of FMF. The regular administration of colchicine at adequate doses is known to improve FMF symptoms in 60–75% of cases [35]. The data obtained from our project are significant in terms of advancing the understanding of the pathogenesis of FMF and contributing to the development of more effective alternative diagnostic and therapeutic methods. Similar studies could help identify new biomarkers and, by screening individuals for these biomarkers, risk profiling could be conducted. This approach could enable the early identification of potential complications in FMF patients, facilitating the implementation of preventive treatments.

## 4. Material and Methods

### 4.1. Study Group

According to the Tel-Hashomer criteria, a total of 30 adult patients clinically diagnosed with FMF were included in this study as the patient group, while 35 healthy adult individuals were included as the control group [36]. Individuals with concomitant cardiovascular, metabolic, renal, or malignancy diseases, or those using medications other than colchicine, were excluded from this study. Peripheral blood samples were collected from patients and healthy controls who signed an informed consent form based on voluntary participation. During physical examinations and blood sampling, patients experiencing active episodes were classified as ’active’, whereas those in attack-free periods were classified as ’inactive’. Clinical data, including detailed examination findings, blood pressure levels, routine biochemical parameters, and the colchicine dosage used by the patients, were collected. This study was approved by the Clinical Research Ethics Committee of Alanya Alaaddin Keykubat University on 26 September 2019, with decision number 10/16.

### 4.2. Total RNA Isolation

Collected peripheral blood samples from patients and healthy controls were supplemented with RNA Save solution (Biological Industries, Beit Haemek Ltd., Kibbutz Beit Haemek, Israel) on the same day for RNA stabilization. Then, total RNA isolation was performed using an RNA purification kit (Norgen Biotek Corp., Canada) for both patients and healthy controls, following the manufacturer’s instructions. The concentration and purity of the isolated total RNA were determined spectrophotometrically using Biotek Synergy H1 Multimode Reader (BioTek Instruments, Inc., US). RNA samples with an A260/A280 ratio between 1.8 and 2.0 were selected for further analysis. Total RNA with optimal purity and sufficient quantity was successfully obtained from 30 patients and 21 healthy controls.

### 4.3. Serum Isolation

Blood samples from patients and controls were collected in gel tubes and allowed to clot at room temperature. Following clotting, the samples were centrifuged at 2000 rpm for 10 min in a refrigerated centrifuge. The resulting serum was aliquoted and stored at −20 °C until the day of analysis.

### 4.4. Quantitative Real-Time Polymerase Chain Reaction (qRT-PCR)

Using an EvoScript Universal cDNA Master kit (Roche, Basel, Switzerland), RNA samples from 30 patients and 21 healthy controls were reverse transcribed to cDNA. Glyceraldehyde-3-phosphate dehydrogenase (*GAPDH*) was used as a reference to quantify and normalize the mRNA expression of *ENHO* (NM_198573.3). The SYBR green-based real-time polymerase chain reaction (RT-PCR) was performed using Light Cycler 96 (Roche Diagnostics, Indianapolis, IN, USA) to assess the amounts of *ENHO* and *GAPDH*. Each sample was analyzed in two replicates. The oligonucleotide primer sequences used for qRT-PCR were as follows: for the *ENHO* —forward 5′-CCATTCTCGCTCTGCCGAC-3′, reverse 5′-CAAGCTGGCTAGACTCTGGG-3′, and for the *GAPDH* gene —forward 5′-GTCTCCTCTGACTTCAACAGCG -3′, reverse 5′-ACCACCCTGTTGCTGTAGCCAA -3′. ΔCt values for both patient and control groups were determined using the formula ΔCt = Ct (Target gene) − Ct (Reference gene), where the Ct values for the target gene, *ENHO*, and the reference gene, *GAPDH*, were substituted accordingly. Relative *ENHO* mRNA expression levels in the patient and control groups were calculated using the 2^−ΔCt^ method.

### 4.5. Enzyme-Linked Immunosorbent Assay (ELISA)

Serum samples from 30 patients and 35 healthy controls were retrieved from the deep freezer on the day of the analysis and thawed at room temperature. The serum Adropin levels of the patients and controls were measured using an Human Adropin ELISA kit (Elabscience, Houston, TX, USA) following the manufacturer’s instructions. A Biotek Synergy H1 Multimode Reader (BioTek Instruments, Inc., Winooski, Vermont, VT, USA) was used as the ELISA reader for analysis.

### 4.6. Statistical Analyses

Statistical analyses were performed using GraphPad Prism 7.0. Normality of continuous variables was assessed with the Shapiro–Wilk test. Group comparisons were conducted using Student’s *t*-test for parametric data and the Mann–Whitney U test for non-parametric data. Patients were sub-grouped based on the presence of particular symptoms (e.g., muscle pain, chest pain, and scrotal pain), and clinical, biochemical, and genetic parameters were compared between groups. These variables were also compared between active and inactive patients. Correlations between clinical features (e.g., age, disease onset, and duration) and laboratory findings (e.g., Adropin levels and *ENHO* expression) were assessed using Pearson’s or Spearman’s correlation analysis, where appropriate. A *p*-value < 0.05 was considered statistically significant.

## 5. Conclusions

This study sheds light on the role of *ENHO* expression and Adropin in the pathophysiology of Familial Mediterranean Fever (FMF). Our findings demonstrate that while *ENHO* expression is significantly reduced in clinically diagnosed FMF patients compared to the controls, the serum Adropin levels remain unchanged between the groups. However, the positive correlations between Adropin levels, patient age, and disease onset point to its potential protective role in FMF. These observations underscore the need for larger cohort studies to validate Adropin as a biomarker and to explore its therapeutic implications. Future investigations should also incorporate genotypic analyses and focus on elucidating the molecular mechanisms underlying the involvement of Adropin or *ENHO* expression in FMF.

## Figures and Tables

**Figure 1 ijms-26-02371-f001:**
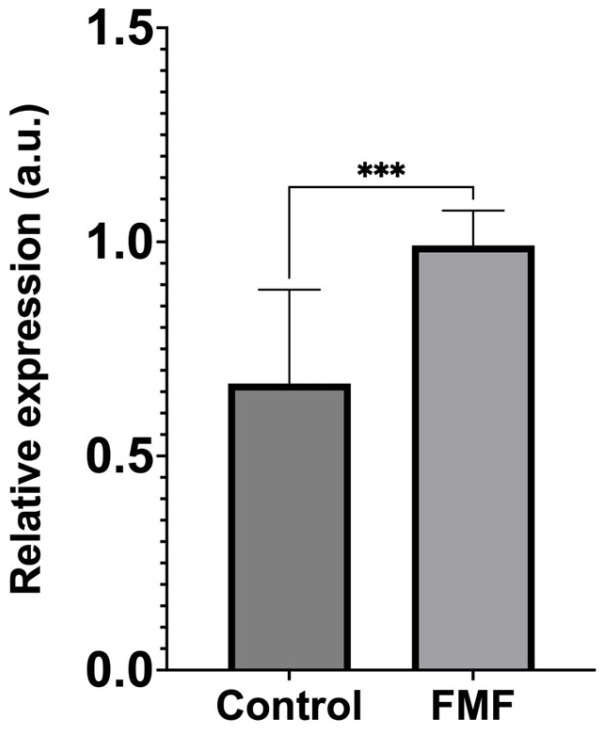
Comparison of relative *ENHO* expression levels between healthy controls and FMF patients. *ENHO* expression levels were significantly higher in the FMF patient group compared to the control group (Mann–Whitney U test, *** *p* = 0.0007). Data are calculated as 2^−ΔCt^ and represented in arbitrary units (a.u.). Error bars indicate standard error of the mean.

**Figure 2 ijms-26-02371-f002:**
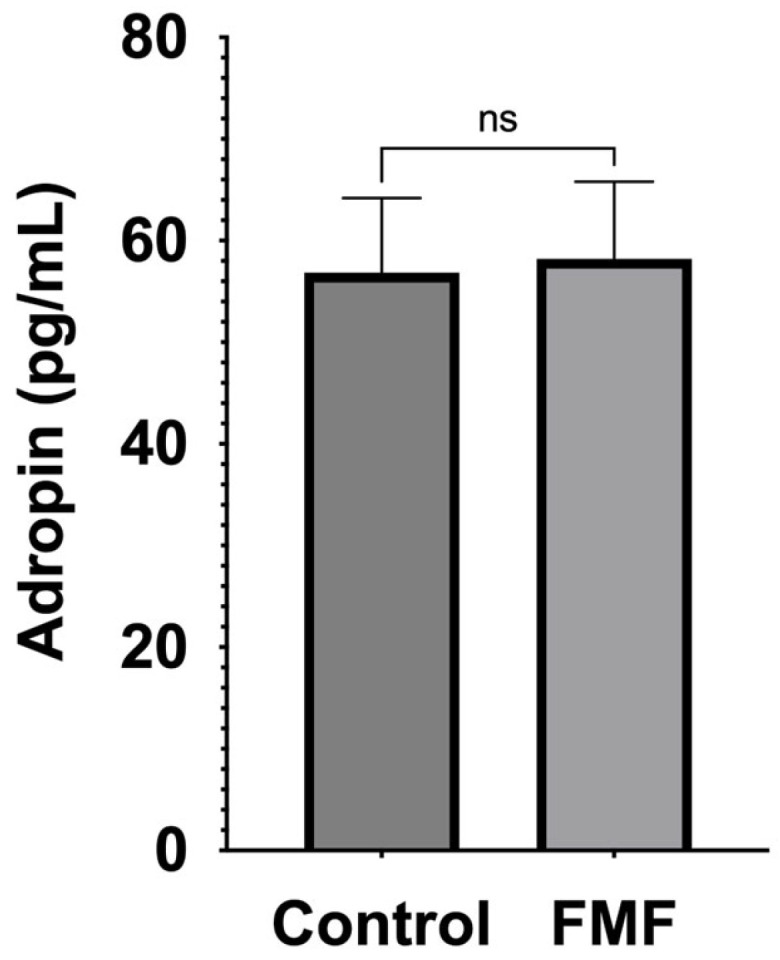
Comparison of serum Adropin levels of control and patient groups. Error bars indicate standard error of the mean. Statistical analysis: Mann–Whitney U test; ns: nonsignificant.

**Figure 3 ijms-26-02371-f003:**
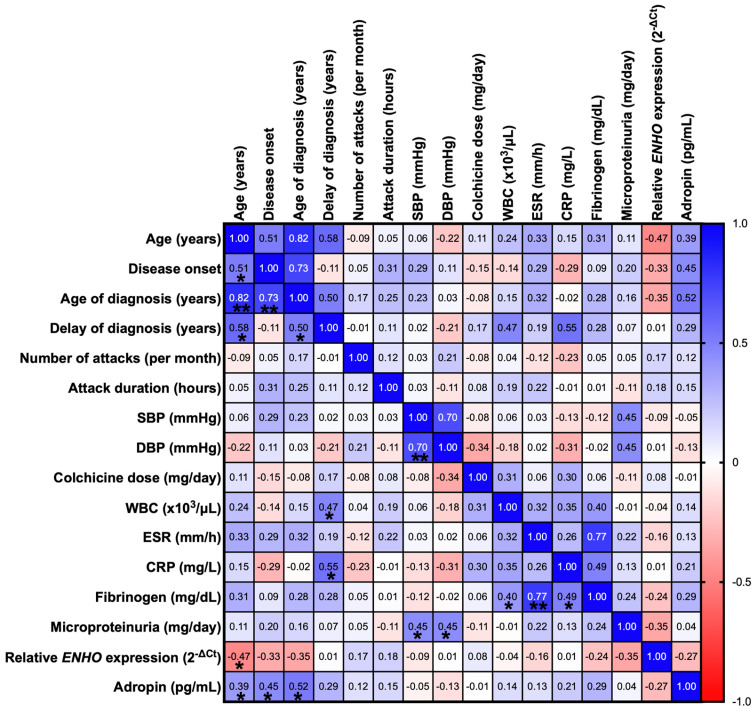
Correlation analysis between the clinical variables and Adropin levels, and *ENHO* expression levels in the FMF patient group. The results are displayed in a correlation matrix. A Spearman’s correlation analysis is applied to assess the relationships between the variables, except for Fibrinogen, relative *ENHO* expression, and age of diagnosis, which follow a Gaussian distribution. A Pearson’s correlation analysis is applied to these variables. Each number within the boxes corresponds to the correlation coefficient between two variables. The cells are color-coded based on the strength and direction of the correlations: shades of blue indicate positive correlations, while shades of red represent negative correlations. Cells marked with an asterisk (*) indicate statistical significance at the level *p* < 0.05; cells with (**) indicate statistical significance at the level *p* < 0.001. SBP: systolic blood pressure; DBP: diastolic blood pressure; WBC: white blood cell; ESR: erythrocyte sedimentation rate; CRP: C-reactive protein.

**Table 1 ijms-26-02371-t001:** Demographic, clinical, and routine biochemical analysis findings, and medication information of patients with FMF.

	Mean (SD)	Min–Max
Age (years)	34.33 (13.79)	19–65
Disease onset (years)	18.45 (10.82)	3–47
Age of diagnosis (years)	27.66 (12.16)	7–52
Delay in diagnosis (years)	9.28 (7.89)	1–25
Number of attacks (per month)	2.10 (2.11)	0–10
Attack duration (hours)	57.28 (24.79)	5–96
Systolic blood pressure	116.7 (8.20)	95–130
Diastolic blood pressure	74.63 (6.19)	65–90
Colchicine dose (mg/day)	0.86 (0.61)	0–2
WBC (×10^3^/μL)	8022 (3516)	4000–18,660
ESR (mm/h)	9.83 (7.70)	2–33
CRP (mg/L)	13.21 (24.37)	2–111
Fibrinogen (mg/dL)	364.1 (100.6)	194–643
Microproteinuria (mg/day)	195.1 (330.3)	7–1742
	w n (%)	w/o n (%)
Fever	15 (50%)	15 (50%)
Abdominal pain	24 (80%)	6 (20%)
Joint pain	22 (73%)	8 (27%)
Chest pain	14 (47%)	16 (53%)
Muscle pain	22 (73%)	8 (27%)
Erysipelas-like erythema	1 (3%)	29 (97%)
Scrotal pain	9 (30%)	21 (70%)

SD: standard deviation; WBC: white blood cell; ESR: erythrocyte sedimentation rate; CRP: C-reactive protein; w: with; w/o: without.

**Table 2 ijms-26-02371-t002:** Comparisons of clinical parameters, *ENHO* expression, and Adropin levels based on patients’ disease symptoms.

	Disease Characteristics																	
Clinical Parameters	Fever			Abdominal Pain			Joint Pain			Chest Pain			Muscle Pain			Scrotal Pain		
	Mean (SD)		*p* *	Mean (SD)		*p* *	Mean (SD)		*p* *	Mean (SD)		*p* *	Mean (SD)		*p* *	Mean (SD)		*p* *
	w	w/o		w	w/o		w	w/o		w	w/o		w	w/o		w	w/o	
Number of attacks (per month)	2.93 (2.79)	1.33 (0.62)	0.05	2.39 (2.271)	1.00 (0.632)	**0.04**	2.41 (2.32)	1.14 (0.69)	0.08	2.60 (2.80)	1.57 (0.76)	0.68	2.38 (2.38)	1.38 (0.92)	0.28	2.56 (2.30)	1.90 (2.05)	0.47
Attack duration (hours)	61.71 (22.50)	53.13 (26.85)	0.48	60.74 (24.39)	44.00 (23.60)	0.17	58.91 (21.88)	52.14 (33.94)	0.75	57.93 (24.54)	56.57 (25.96)	0.85	57.14 (23.36)	57.63 (29.98)	0.87	51.22 (29.45)	60.00 (22.70)	0.47
SBP (mmHg)	117.31 (5.99)	116.07 (10.03)	0.72	117.50 (6.86)	113.00 (13.04)	0.45	116.00 (8.68)	118.57 (6.90)	0.50	117.67 (6.78)	115.42 (9.88)	0.58	115.79 (8.21)	118.75 (8.35)	0.52	114.44 (9.82)	117.78 (7.32)	0.49
DBP (mmHg)	75.77 (4.94)	73.57 (7.19)	0.28	76.14 (5.76)	68.00 (2.74)	**0.01**	75.00 (6.28)	73.57 (6.27)	0.69	77.00 (5.92)	71.67 (5.37)	**0.04**	75.00 (6.67)	73.75 (5.18)	0.74	75.56 (7.68)	74.17 (5.49)	0.78
Colchicine dose (mg/day)	0.71 (0.61)	1.00 (0.60)	0.31	0.76 (0.60)	1.25 (0.52)	0.10	0.95 (0.60)	0.57 (0.61)	0.20	0.93 (0.59)	0.79 (0.64)	0.53	0.86 (0.59)	0.88 (0.69)	0.99	0.78 (0.51)	0.90 (0.66)	0.63
WBC (x10^3^/μL)	8185.33 (3415.94)	7858 (3724.93)	0.44	7719.58 (3339.23)	9230.00 (4266.27)	0.37	8290.87 (3887.45)	7137.14 (1770.25)	0.67	8111.33 (3921.50)	7932.00 (3194.65)	0.87	8658.18 (3804.83)	6271.25 (1731.22)	0.97	7226.67 (3565.23)	8362.38 (3525.65)	0.26
ESR (mm/h)	9.29 (6.01)	10.33 (9.19)	0.88	9.04 (6.40)	12.83 (11.75)	0.48	9.36 (6.89)	11.29 (10.36)	0.78	9.36 (6.46)	10.27(8.91)	0.88	11.00 (8.40)	6.75 (4.53)	0.28	5.25 (4.03)	11.57 (8.11)	**0.04**
CRP (mg/L)	10.15 (16.39)	16.27 (30.68)	0.74	10.18 (17.63)	25.33 (42.35)	0.13	13.80 (27.62)	11.29 (8.26)	0.21	13.62 (21.51)	12.80 (27.70)	0.35	15.73 (28.04)	6.29 (5.18)	0.84	22.37 (38.97)	9.29 (14.14)	0.72
Fibrinogen (mg/dL)	389.79 (101.90)	340.20 (96.48)	0.22	363.35 (104.35 )	367.17 (93.26)	0.81	362.91 (108.03)	368.00 (79.43)	0.75	382.36 (119.84)	347.13 (79.02)	0.43	382.33 (111.04)	316.38 (40.15)	0.17	342.75 (100.97)	372.29 (101.65)	0.58
Microproteinuria (mg/day)	139.54 (120.83)	246.64 (445.79)	0.91	219.64 (362.20)	87.00 (48.30)	0.56	213.25 (378.19)	143.14 (124.53)	0.65	297.57 (439.74)	84.69 (37.30)	**0.03**	224.35 (373.85)	111.43 (138.11)	0.48	168.25 (169.07)	206.37 (382.08)	0.66
Relative *ENHO* expression (2^−ΔCt^)	0.00091 (0.00034)	0.00107 (0.00054)	0.60	0.001 (0.00043)	0.00097 (0.00056)	0.86	0.001 (0.00043)	0.00088 (0.00052)	0.36	0.00105 (0.0004)	0.00093 (0.0005)	0.37	0.00091 (0.00044)	0.00122 (0.00043)	0.06	0.00104 (0.00044)	0.00097 (0.00046)	0.69
Adropin levels (pg/mL)	56.34 (37.21)	60.10 (46.66)	0.87	60.77 (44.57)	48.02 (26.47)	0.71	61.83 (45.23)	46.34 (24.79)	0.56	59.44 (49.13)	57.00 (33.96)	0.54	66.57 (44.80)	35.24 (17.16)	0.50	58.33 (47.71)	58.17 (39.85)	0.72

* Mann–Whitney U test; bold values indicate statistical significance. SD: standard deviation; w: with; w/o: without; SBP: systolic blood pressure; DBP: diastolic blood pressure; WBC: white blood cell; ESR: erythrocyte sedimentation rate; CRP: C-reactive protein.

**Table 3 ijms-26-02371-t003:** Comparison of clinical parameters, *ENHO* expression, and Adropin levels based on patients’ attack status.

	Attack Status of Patients		
Parameters	Active (n = 6) Mean (SD)	Inactive (n = 24) Mean (SD)	*p* *
SBP (mmHg)	110.83 (11.14)	118.33 (6.59)	0.07
DBP (mmHg)	68.33 (2.59)	76.43 (5.73)	**0.004**
WBC (×10^3^/μL)	9793.33 (4034.13)	7578.75 (3319.93)	0.14
ESR (mm/h)	15.50 (11.22)	8.35 (5.99)	0.14
CRP (mg/L)	47.83 (39.44)	4.56 (4.45)	**<0.001**
Fibrinogen (mg/dL)	430.50 (90.31)	346.83 (97.47)	**0.036**
Microproteinuria (mg/day)	97.60 (21.09)	217.22 (363.58)	0.56
Relative *ENHO* expression (2^−ΔCt^)	0.00083 (0.00041)	0.00103 (0.00046)	0.35
Adropin levels (pg/mL)	66.97 (40.59)	56.02 (42.30)	0.49

* Mann–Whitney U test; bold values indicate statistical significance. SD: standard deviation; SBP: systolic blood pressure; DBP: diastolic blood pressure; WBC: white blood cell; ESR: erythrocyte sedimentation rate; CRP: C-reactive protein.

## Data Availability

The data supporting the conclusions of this article will be made available by the authors on request. The data are not publicly available due to legal issues.

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
