# Peer review of "The Role of Energy Homeostasis-Associated Gene Expression and Serum Adropin Levels in Patients with Familial Mediterranean Fever"

_ijms, 2025, doi:10.3390/ijms26052371_

Round 1
Reviewer 1 Report
Comments and Suggestions for Authors
In this article, Durkadin Demir Eksi et al. present a study on ENHO gene expression and serum Adropin levels in 30 FMF patients. The topic is novel in the context of FMF and highly intriguing. However, I have two major concerns regarding the study design and data interpretation. Please find my detailed comments below:
Major Comments:
1- Classification of FMF Patients:
The primary concern in this study, apart from the small sample size (n = 30), is the classification of FMF patients. The authors mention using the Tel-Hashomer criteria for patient inclusion, without incorporating genetic classification. This approach presents a significant limitation. The Tel-Hashomer criteria have several weaknesses, particularly in terms of sensitivity and specificity, applicability to atypical presentations, overlapping symptoms with other autoinflammatory diseases, and, most importantly, the absence of genetic testing.
Given these limitations, the accuracy of the FMF diagnosis in this cohort is questionable. I strongly recommend sequencing the MEFV gene for all 30 patients to confirm their diagnosis, assess genotype-phenotype correlations, and evaluate symptom severity. This is especially pertinent as emerging research indicates that FMF phenotype severity is influenced by genotype. For example, a recent large-scale genetic study in Lebanon demonstrated that homozygous M694V and V726A patients exhibited more severe phenotypes compared to E148Q carriers (doi: 10.3389/fgene.2024.1506656). Incorporating this genetic information would not only enhance diagnostic accuracy but also provide an opportunity for more robust statistical analyses, potentially identifying correlations between MEFV genotypes, and ENHO gene expression, or serum Adropin levels.
2- Interpretation of qRT-PCR Results:
I have a major concern regarding the interpretation of the qRT-PCR results. In Figure 1, the authors present the ΔCt values for controls and FMF patients, showing a clear and significant difference: ΔCt values are lower in the FMF group. In qPCR, a lower ΔCt typically signifies higher relative expression or abundance of the target gene/transcript.
However, the authors claim in the figure legend and throughout the manuscript (e.g., results section, lines 178–179) that “ENHO gene expression levels were significantly higher in the control group compared to FMF patients.” This directly contradicts the data shown in Figure 1, which instead suggests that FMF patients exhibit higher ENHO gene expression.
This misinterpretation is a critical issue and must be addressed. The authors need to revise this statement throughout the manuscript and ensure that the results and discussion sections align with the actual findings in Figure 1.
Furthermore, I recommend presenting gene expression levels as 2^-ΔCt in arbitrary units (A.U.) instead of ΔCt values. This approach provides a more intuitive visualization of expression levels and directly reflects the relative expression of the target gene.
Minor comments:
- Page 2 line 52-54: what about ENHO expression in blood, and particularly PBMCs? if it is not expressed in blood, then the entire study is not feasible.
- Page 2, line 61-62: but, a high proportion of FMF show complete/partial response to colchicine, and around 5-10% show no response. Please rephrase this sentence.
- Page 2, line 64-65: is it really true that 30-40% of FMF patients do not carry a mutation in the MEFV gene? However, in the Discussion section, the authors state that this number is of around 50% (page 8, line 162). Are the authors referring to FMF patients only? or all patients with periodic fevers in general? Please clarify. Also, please recheck this discrepancy and the accuracy of both numbers. To my knowledge, the number is lower than 50%, unless the authors are referring to all periodic fevers.
- Page 3, table 1: amyloidosis is not reported among the clinical manifestations. It was not tested? Or none of the patients presented it? It would be worth mentioning.
- Page 4, line 112: … were also found IN patients with …
- Page 9, paragraph 2: the authors say that “clinical studies have revealed reduced adropin expression in inflammatory diseases…” (line 176-178); however, in the following sections of the paragraph they say “increase” (line 184), “elevated” (line 187). Which information is correct? Please recheck the literature and correct accordingly.
- Page 9, line 193-195: this explanation provided by authors do not fit with their observation of “lower ENHO expression in FMF patients” ; however it fits with the correct “higher” expression in FMF patients.
- End of the discussion and conclusion: the authors acknowledge the limitations of their study, but these limitations are considered a major issue.
- I suggest moving the “Material and Methods” to section 2, after the introduction. Of course, according to the journal’s recommendations.
- Page 11, section 4.2: I suggest splitting RNA isolation and serum isolation for more clarity.
- Page 4.3, section 4.3: did the authors run the analysis in duplicated or triplicates? This information should be added.
Reviewer 2 Report
Comments and Suggestions for Authors
In this article, Durkadin Demir Eksi et al. compared the gene expression level of ENHO and the serum level of its protein Adropin between 30 Familial Mediterranean Fever (FMF) patients and 35 healthy control, with qRT-PCR and ELISA respectively. Lower ENHO levels were found in patients, but the differences of the protein level are not significant. Subsequential analysis identified potential association between the expression of ENHO, Adropin levels, and the age of patients. While MEFV gene mutation is the major reason and diagnosis standard for FMF, some non-typical patients could carry no mutations in MEFV. Adropin has a role in inflammation, while FMF is a disease related to autoinflammation, they may have a potential link but has not been exclusively studied. From this aspect, this work has an innovation, which can be potentially published in the International Journal of Molecular Sciences. In general, this manuscript is well documented and discussed for the limitations. However, the quality of the data needs to be improved significantly. With the following concerns, the conclusion and potential correlation between FMF and ENHO gene expression is not well supported.
1. In consistency were present for the sample numbers. For instance, 35 healthy controls were used for ELISA test, while only 21 controls were used for RT-qPCR. What’s the criteria used to exclude the other 1/3 of the samples? Is a lower RNA quality or quantity the only reason for selecting a smaller sample population for comparing the gene expression levels? Other than the inconsistency in techniques, is there any physiological reason causing the quality issue for some of these patients? As the authors mentioned in the discussion section, the reason for a significant reduction in mRNA levels but not protein levels could be because the qPCR did not survey all the control samples. Will that bias cause unreliable differences that could be caught if all the samples were assessed? Also, is there any publication that can support a difference in post-translational degradation of this gene under different physiological conditions that may cause the observed diversity?
2. In the result section, it says that the 30 patients have a mean age of 33.05±13.3 years, while Table 1 that summarized more details of these patients says that the patients were at the age between 19-65, with a mean of 34.33 and a SD of 13.79. The two sets of statistics provided about mean ages are close but not identical. This discrepancy suggests that they may have been derived from different datasets or calculations. To ensure clarity and accuracy, please confirm whether these values pertain to the same cohort and, if so, verify the calculations to maintain consistency. Furthermore, I recommend using the format 'mean (SD)' instead of 'mean ± SD' for reporting, as this aligns with common style guidelines. If these statistics represent different cohorts or analyses, they should be clearly distinguished in the text to avoid confusion.
3. Why were the Mann-Whitney U test and Spearman correlation analysis chosen for the statistical analysis? These tests are typically used when data do not follow a normal distribution. Can you confirm whether this assumption holds true for both datasets?
4. To improve clarity, please explain the definition of active and in active in “Six out of 30 patients (20%) were active, and the rest were inactive.” in the place that they are first described instead of explaining in the discussion section. Additionally, please explain “positive correlation between disease onset and adropin levels in FMF patients” What is meant by a positive correlation in this context? Does a higher adropin level correspond to a later onset of the disease?
5. The formatting of proteins and genes should be consistent and accurate throughout the manuscript. For instance, in the abstract, the titles of Figure 2 and Table 2, and some places in the main text, “adropin levels” were used. It should be revised to “Adropin levels” to align with the standard conventions for protein names.
6. “Calculation of the 2-∆∆Ct value indicated that ENHO expression in 89 the control group was elevated by 3.72-fold relative to the patient group.” Normally people compare how patients changed when compared with control group. Therefore, it should be 3.72-fold decrease in the patient group in comparison with control group.
7. In figure 1, the error bar for qPCR is typically SD or SEM. Is that standard to show minimum and maximum values?
8. No sequencing results for MEFV gene mutations is acknowledged in the discussion section. In future study that expand the sample number, this could be an important factor to consider, since MEFV gene is a major genetic factor, which may complicate the analysis for the association between FMF and ENHO expressions.
Reviewer 3 Report
Comments and Suggestions for Authors
The paper is interesting and well written. The authors investigated the relationship between Energy Homeostasis-Associated Gene (ENHO) gene expression, adropin levels, and Familial Mediterranean Fever (FMF), examining their correlations with disease characteristics. The study confirmed that ENHO downregulation is involved in FMF pathogenesis, while the correlation between adropin levels and disease onset indicates a potential protective role. I suggest to discuss the link with Th17 cells, IL-31/IL-33 axis and microbiome that impact on immune responses and inflammations (see and add as references papers by Murdaca et al concerning Th17 cells, IL-31/IL-33 axis and microbiome and immune response).
Comments on the Quality of English LanguageMinor english editing
Round 2
Reviewer 1 Report
Comments and Suggestions for Authors
I Acknowledge the efforts made by the authors to address my concerns and correct the misinterpretation of qPCR data.
Concerning genetic testing in the context of FMF:
I thank the authors for the explanation of the FMF clinical classification. But, without genetic testing of the MEFV gene, a definitive diagnosis of Familial Mediterranean Fever (FMF) cannot be confirmed at the molecular level. However, FMF can be diagnosed clinically using the Tel-Hashomer criteria, especially in regions where the disease is prevalent, like in Turkey. Nevertheless, since the diagnosis is based only on clinical criteria, it is more precise to say: “patients clinically-diagnosed with FMF”
Without genetic confirmation, it is better to avoid using "FMF patient", as other autoinflammatory conditions can mimic FMF. And, the precise diagnosis can always be questionable.
So, please use the term “clinically-FMF” in the text.
Comment: The authors acknowledge the limitation in their study, as no genetic testing was done. However, I still find it weird for patient management, since the symptoms may vary according to the genetic mutation found. And importantly, the inflammatory signs and severity are reportedly different. And, also find it weird for genetic counselling for FMF patients and their families.
The authors are invited to acknowledge clearly this limitation in the discussion (absence of genetic testing), line 317-317; and state in a straight-forward manner that genetic testing is important for a better management of the disease and symptoms severity, and also genetic counselling. Please add adequate references (i.e. PMID: 32824452 and 39897620).
Concerning qPCR data:
The authors have corrected the errors done in the previous analysis, and corrected the values and adjusted the interpretation. Now, the corrected conclusions are comparable to the literature.
Minor comments:
The authors have adequately addressed the minor comments.
